# SELECTIVE FREQUENCY NETWORK FOR IMAGE RESTORATION

**Yuning Cui[1], Yi Tao[2], Zhenshan Bing[1], Wenqi Ren**[*3,5], **Xinwei Gao[4], Xiaochun Cao**[3,5], **Kai Huang[3], Alois Knoll[1]**

[1]Technical University of Munich  [2]MIT Universal Village Program  [3]Sun Yat-sen University
[4]Tencent  [5]Chinese Academy of Sciences
{yuning.cui,bing,knoll}@in.tum.de,yitao@universal-village.org,
{renwq3,caoxiaochun,huangk36}@mail.sysu.edu.cn,vitogao@tencent.com

## ABSTRACT

Image restoration aims to reconstruct the latent sharp image from its corrupted counterpart. Besides dealing with this long-standing task in the spatial domain, a few approaches seek solutions in the frequency domain in consideration of the large discrepancy between spectra of sharp/degraded image pairs. However, these works commonly utilize transformation tools, *e.g.*, wavelet transform, to split features into several frequency parts, which is not flexible enough to select the most informative frequency component to recover. In this paper, we exploit a multi-branch and content-aware module to decompose features into separate frequency subbands dynamically and locally, and then accentuate the useful ones via channel-wise attention weights. In addition, to handle large-scale degradation blurs, we propose an extremely simple decoupling and modulation module to enlarge the receptive field via global and window-based average pooling. Integrating two developed modules into a U-Net backbone, the proposed **S**elective **F**requency **Net**work (SFNet) performs favorably against state-of-the-art algorithms on five image restoration tasks, including single-image defocus deblurring, image dehazing, image motion deblurring, image desnowing, and image deraining [1].

## 1 INTRODUCTION

Image restoration aims to recover a high-quality image by removing degradations, *e.g.*, noise, blur, and snowflake. In view of its important role in surveillance, self-driving techniques, and remote sensing, image restoration has gathered considerable attention from industrial and academic communities. However, due to its ill-posed property, many conventional approaches address this problem based on various assumptions (Zhang et al., 2022; Yang et al., 2020b) or hand-crafted features (Karaali & Jung, 2017), which are incapable of generating faithful results in real-world scenarios.

Recently, deep neural networks have witnessed the rapid development of image restoration and obtained favorable performance compared to conventional methods. A flurry of convolutional neural networks (CNN) based methods have been developed for diverse image restoration tasks by inventing or borrowing advanced modules, including dilated convolution (Luo et al., 2022; Zou et al., 2021), U-Net (Ronneberger et al., 2015), residual learning (Zhang et al., 2017), multi-stage pipeline (Zhang et al., 2019b), and attention mechanisms (Liu et al., 2019). However, with convolution units, these methods have limited receptive fields, and thus they are not capable of capturing long-range dependencies. This requirement is essential for restoration tasks, since a single pixel needs information from its surrounding region to be recovered. More recently, many researchers have tailored Transformer (Vaswani et al., 2017) for image restoration tasks, such as motion deblurring (Tsai et al., 2022), dehazing (Guo et al., 2022; Song et al., 2022) and desnowing (Chen et al., 2022b;c).

Nonetheless, the above-mentioned methods mainly conduct restoration in the spatial domain, which do not sufficiently leverage frequency discrepancies between sharp/degraded image pairs. To this end, a few works utilize the transformation tools, *e.g.,* wavelet transform or Fourier transform, to

---

*Corresponding author
[1]Our code and models are available at `https://github.com/c-yn/SFNet`.

decompose features into different frequency components and then treat separate parts individually to reconstruct the corresponding feature (Selesnick et al., 2005; Yang & Fu, 2019; Zou et al., 2021; Mao et al., 2021). Nevertheless, wavelet transform decouples the feature map into different subbands in a fixed manner, and thus it is not capable of distinguishing the most informative or useless frequency components to enhance or suppress. In addition, these methods need corresponding inverse Fourier/wavelet transform, leading to additional computation overhead.

To overcome the above drawbacks and select the most informative frequency component to reconstruct, we propose a novel decoupling and recalibration module for image restoration tasks, named Multi-branch Dynamic Selective Frequency module (MDSF). Specifically, we utilize the multi-branch learnable filters to generate high- and low-frequency maps dynamically and locally, and then leverage the channel-wise attention mechanism, modified from (Li et al., 2019), to emphasize or suppress the resulting frequency components. Our module has two key advantages. Firstly, according to the input and task, the decoupling step dynamically generates filters to decompose feature maps. Secondly, our module does not introduce extra inverse transform.

Receptive field is another critical factor for image restoration tasks due to the various sizes of degradation blurs (Suin et al., 2020; Son et al., 2021). To complement the above dynamic module, MDSF, that processes features locally, we further propose a simple yet effective module, dubbed Multi-branch Compact Selective Frequency module (MCSF), to enhance the helpful frequency signals based on multiple and relatively global receptive fields. Specifically, we utilize global and window-based average pooling techniques to attain disparate frequency maps, and then use learnable parameters to modulate the resulting maps without resorting to any convolution layers. Compared to MDSF, besides the enlarged receptive fields, MCSF is lightweight enough to be embedded in multiple positions of the backbone. The main contributions of this study are summarized as follows:

- We propose a multi-branch dynamic selective frequency module (MDSF) that is capable of decoupling feature maps into different frequency components dynamically via the theoretically proved filters, and selecting the most informative components to recover.
- We develop a multi-branch compact selective frequency module (MCSF) that performs frequency decoupling and recalibration using multi-scale average pooling operations to pursue a large receptive field for large-scale degradation blurs.
- Incorporating MDSF and MCSF into a U-shaped backbone, the proposed selective frequency network (SFNet) achieves state-of-the-art results on five image restoration tasks, including image defocus/motion deblurring, dehazing, deraining, and desnowing.

## 2 RELATED WORK

**Image Restoration.** Prior to the deep learning era, a great number of methods have been proposed for image restoration problems based on various assumptions and hand-crafted features (Sezan & Tekalp, 1990; Kundur & Hatzinakos, 1996; Calvetti et al., 1999). In recent years, with the rapid development of deep learning, a flurry of approaches have been investigated utilizing convolutional neural networks for image motion deblurring (Zamir et al., 2021; Yuan et al., 2020; Cui et al., 2023; Purohit et al., 2021), defocus deblurring (Ruan et al., 2022; Abuolaim & Brown, 2020; Son et al., 2021), desnowing (Chen et al., 2021c; 2020a), dehazing (Dong et al., 2020; Liu et al., 2019; Ren et al., 2016; Zhang et al., 2018), and deraining (Yang et al., 2020b; Wang et al., 2019).

More recently, to capture long-range dependencies, many works have borrowed Transformer (Vaswani et al., 2017) from the natural language processing field into image restoration (Chen et al., 2021a; Liang et al., 2021; Zamir et al., 2022; Wang et al., 2022) and specific tasks such as image motion deblurring (Tsai et al., 2022), dehazing (Song et al., 2022), and desnowing (Chen et al., 2022b). In this study, instead of exploiting a more advanced backbone for image restoration, we pay more attention to the frequency selection mechanism based on efficient CNN.

**Frequency Based Image Restoration.** Many algorithms have been developed to address various low-level vision problems from a frequency perspective. Specifically, Chen *et al.* (Chen et al., 2021c) propose a hierarchical desnowing network based on dual-tree complex wavelet representation (Selesnick et al., 2005). Yang *et al.* (Yang & Fu, 2019) develop the wavelet based U-Net to replace up-sampling and down-sampling. Zou *et al.* (Zou et al., 2021) utilize wavelet transform based module to help recover texture details. Yang *et al.* (Yang et al., 2020a) devise a wavelet structure similarity loss function for training. Mao *et al.* (Mao et al., 2021) use Fourier transform to

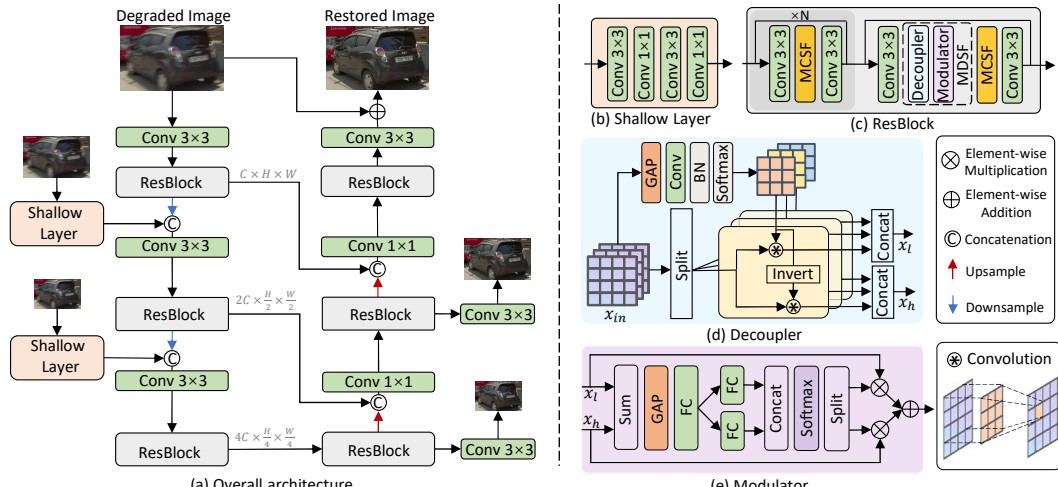

Figure 1: (a) Overall architecture of the proposed SFNet. (b) Shallow layer extracts the shallow feature for low-resolution images. (c) ResBlock contains the proposed modules: MDSF (Decoupler (d) and Modulator (e)) and MCSF. MDSF is shown in the one-branch case for clarity. *Invert* depicts the operation of subtracting the low-pass filter from the identity filter.

integrate both high- and low-frequency residual information. Yoo *et al.* (Yoo et al., 2018) complete image restoration based on the estimation of the DCT coefficient distribution. In this work, we pursue a dynamic and efficient manner to select the useful frequency part at multiple receptive fields.

## 3 METHODOLOGY

We first describe the overall architecture of SFNet (Fig. 1 (a)). Then we present the proposed modules, MDSF (Fig. 1 (d,e)) and MCSF. The loss functions follow in the final part.

### 3.1 OVERALL ARCHITECTURE

Our network adopts the encoder-decoder architecture to learn hierarchical representations. Specifically, SFNet consists of a three-scale decoder and a three-scale encoder. Each scale is comprised of a ResBlock (Fig. 1 (c)). MDSF is only deployed in the last residual block of each ResBlock while MCSF exists in all blocks. Following previous methods (Cho et al., 2021; Mao et al., 2021; Tu et al., 2022), multi-input and multi-output mechanisms are used to ease training difficulty. Specifically, input images of reduced sizes are merged into the main path via the shallow layer (Fig. 1 (b)), and the predicted images are produced by $3 \times 3$ convolutional layers after each scale of decoder. In addition, we adopt feature-level and image-level skip connections to assist training. In Fig. 1, we only show the top-level image skip connection for clarity. The up-sampling and down-sampling layers are implemented by transposed and strided convolutions, respectively.

### 3.2 MULTI-BRANCH DYNAMIC SELECTIVE FREQUENCY MODULE (MDSF)

To select the informative frequency component to reconstruct, MDSF mainly contains two elements: frequency decoupler (Fig. 1 (d)) and modulator (Fig. 1 (e)). Decoupler decomposes features into separate frequency parts dynamically based on learned filters, and then modulator utilizes channel-wise attention to accentuate the useful frequency. Additionally, to provide various local receptive fields, MDSF splits features among the channel dimension, and then applies different filter sizes to separate parts. We only show the one-branch case in Fig. 1 (d) for simplicity.

To dynamically decompose feature maps, we utilize the learnable and theoretically proven low-pass filter (refer to Appendix B for the proof) and the corresponding high-pass filter to generate low- and high-frequency maps. The learned filters are shared across the group dimension to strike a balance between complexity and feature diversity. Specifically, given any feature map $X \in \mathbb{R}^{C \times H \times W}$, where $C$ is the number of channels and $H \times W$ denotes the spatial dimension, we firstly leverage the filter-generating layer to produce the low-pass filter for each group of the input, formulated as

$$F^l = \text{Softmax}((\mathcal{B}(W(\text{GAP}(X))))) \tag{1}$$

where $F^l \in \mathbb{R}^{k^2 g \times 1 \times 1}$, $k \times k$ is the kernel size of low-pass filter; $g$ denotes the number of groups; $\mathcal{B}, W$, and GAP are Batch Normalization (Ioffe & Szegedy, 2015), the parameters of convolution and global average pooling, respectively. The group-based operation has fewer parameters and lower complexity than generating filters for each pixel. The number of groups is discussed in Sec 4.3.

To attain the high-pass filter, we subtract the resulting low-pass filter from the identity kernel with central value as one and everywhere else as zero. Next, for each group feature $X_i \in \mathbb{R}^{C_i \times H \times W}$, where $i$ is the group index and $C_i = \frac{C}{g}$, its low- and high-frequency components can be obtained by using the corresponding reshaped filter $F^L$ and $F^H$ ($\in \mathbb{R}^{g \times k \times k}$), which is expressed as:

$$X^l_{i,c,h,w} = \sum_{p,q} F^L_{i,p,q} X_{i,c,h+p,w+q}; \quad X^h_{i,c,h,w} = \sum_{p,q} F^H_{i,p,q} X_{i,c,h+p,w+q} \tag{2}$$

where $c$ is the index of a channel; $h$ and $w$ denote spatial coordinates; and $p, q \in \{-1, 0, 1\}$.

After decoupling the feature map into different frequency components, we leverage the frequency modulator to emphasize the genuinely useful part for reconstruction, as illustrated in Fig. 1 (e). The modulator works among the channel dimension based on the modified SKNet (Li et al., 2020). Formally, given two frequency maps, $X^l$ and $X^h$, we first generate the fused feature by,

$$Z = W_{fc}(\text{GAP}(X^l + X^h)) \tag{3}$$

where $W_{fc}$ is the parameters of a fully connected layer. To attain channel-wise weights, we use two other fully connected layers followed by concatenation and Softmax function, formulated as:

$$[W^l, W^h]_c = \frac{e^{[W_l(Z), W_h(Z)]_c}}{\sum_j^{2C} e^{[W_l(Z), W_h(Z)]_j}} \tag{4}$$

where $W^l$ and $W^h$ are channel-wise attention weights for two frequency parts; $W_l$ and $W_h$ are parameters of fully connected layers; $[\cdot, \cdot]$ denotes concatenation; and $c$ is the channel index of concatenated features. Compared to SKNet (Li et al., 2020), which performs Softmax on each channel as $W^l_c = \frac{e^{W_l(Z)_c}}{e^{W_l(Z)_c} + e^{W_h(Z)_c}}$, we consider all channels into consideration to facilitate interactions between different channels of two maps. Then, the final weights can be obtained by split operation.

Based on the above one-branch case, the multiple branches with varied filter sizes can be express as:

$$\hat{X} = [\mathcal{M}_1(\mathcal{D}_1(X_1)), ..., \mathcal{M}_m(\mathcal{D}_m(X_m))] \tag{5}$$

where $\mathcal{D}$ and $\mathcal{M}$ denote decoupler and modulator, respectively, and $X_m$ is the equally split feature.

## 3.3 MULTI-BRANCH COMPACT SELECTIVE FREQUENCY MODULE (MCSF)

Since receptive field plays a critical role in image restoration, where degradation blurs always differ in size (Son et al., 2021; Mao et al., 2021), we develop MCSF to efficiently enlarge the receptive field of SFNet. MCSF has two branches with different receptive fields, $i.e.,$ the global branch and window-based branch. Considering these branches share a similar paradigm, we only detail the window-based one, which is inspired by the idea of window-based attention (Liu et al., 2021).

Specifically, given the split feature $X \in \mathbb{R}^{\frac{C}{2} \times H \times W}$, it is partitioned into four windows, each with the size of $2C \times \frac{H}{2} \times \frac{W}{2}$. To get the low-frequency part, global average pooling is applied to the resulting windows (refer to Appendix C for analyses of this option). The corresponding high-frequency part can be obtained by subtracting the low-frequency map from the partitioned feature. To select the useful frequency subbands, we rescale these two maps by learnable weights, which are directly optimized by backpropagation. Finally, the updated frequency maps are reversed to the original resolution. The global branch has a similar pipeline, yet with a global receptive field.

Compared to MDSF, besides the enlarged receptive field, MCSF does not accomplish frequency decoupling and modulating with the aid of convolution layers, resulting in fewer parameters and lower complexity (see Tab. 9 for details). Hence, MCSF can be embedded in multiple positions.

## 3.4 LOSS FUNCTION

To facilitate the frequency selection process, we adopt L$_1$ loss in both spatial and frequency domains:

$$L_{spatial} = \sum_{r=1}^{3} \frac{1}{S_r} \|\hat{X}_r - Y_r\|_1; \quad L_{frequency} = \sum_{r=1}^{3} \frac{1}{S_r} \|\mathcal{F}(\hat{X}_r) - \mathcal{F}(Y_r)\|_1 \tag{6}$$

Table 1: Quantitative comparisons with previous leading single-image defocus deblurring methods on DPDD testset (Abuolaim & Brown, 2020), which contains 39 outdoor and 37 indoor scenes.

| Method | Indoor Scenes | | | | Outdoor Scenes | | | | Combined | | | |
|---|---|---|---|---|---|---|---|---|---|---|---|---|
| | PSNR↑ | SSIM↑ | MAE↓ | LPIPS↓ | PSNR↑ | SSIM↑ | MAE↓ | LPIPS↓ | PSNR↑ | SSIM↑ | MAE↓ | LPIPS↓ |
| EBDB | 25.77 | 0.772 | 0.040 | 0.297 | 21.25 | 0.599 | 0.058 | 0.373 | 23.45 | 0.683 | 0.049 | 0.336 |
| DMENet | 25.50 | 0.788 | 0.038 | 0.298 | 21.43 | 0.644 | 0.063 | 0.397 | 23.41 | 0.714 | 0.051 | 0.349 |
| JNB | 26.73 | 0.828 | 0.031 | 0.273 | 21.10 | 0.608 | 0.064 | 0.355 | 23.84 | 0.715 | 0.048 | 0.315 |
| DPDNet | 26.54 | 0.816 | 0.031 | 0.239 | 22.25 | 0.682 | 0.056 | 0.313 | 24.34 | 0.747 | 0.044 | 0.277 |
| KPAC | 27.97 | 0.852 | 0.026 | 0.182 | 22.62 | 0.701 | 0.053 | 0.269 | 25.22 | 0.774 | 0.040 | 0.227 |
| IFAN | 28.11 | 0.861 | 0.026 | 0.179 | 22.76 | 0.720 | 0.052 | 0.254 | 25.37 | 0.789 | 0.039 | 0.217 |
| DRBNet | | | | - | | | | | 25.72 | 0.791 | - | 0.183 |
| Restormer | 28.87 | **0.882** | 0.025 | **0.145** | 23.24 | 0.743 | 0.050 | **0.209** | 25.98 | **0.811** | 0.038 | **0.178** |
| **SFNet** | **29.16** | 0.878 | **0.023** | 0.168 | **23.45** | **0.747** | **0.049** | 0.244 | **26.23** | 0.811 | **0.037** | 0.207 |

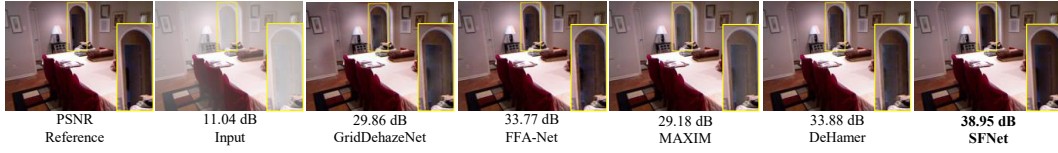

Figure 2: Single-image defocus deblurring results on the DPDD dataset (Abuolaim & Brown, 2020).

where $r$ denotes the index of input/output images of different resolutions; $\mathcal{F}$ represents fast Fourier transform; $S_r$ is the number of elements for normalization; and $\hat{X}_r, Y_r$ are output and target images, respectively. The final loss function is given by $L = L_{spatial} + \lambda L_{frequency}$, where $\lambda$ is set as 0.1.

# 4 EXPERIMENTAL RESULTS

## 4.1 SETTINGS

We evaluate the proposed SFNet on five restoration tasks: image motion/defocus deblurring, image deraining, image dehazing, and image desnowing. More details of the used datasets and training settings for each task are provided in Appendix A. FLOPs are computed on patch size of $256 \times 256$.

We train separate models for different tasks. Unless mentioned otherwise, the following parameters are adopted. The batch size is set as 4 with patch size of $256 \times 256$. Each patch is randomly flipped horizontally for data augmentation. The initial learning rate is $1e^{-4}$ and gradually reduced to $1e^{-6}$ with the cosine annealing (Loshchilov & Hutter, 2016). Adam ($\beta_1 = 0.9, \beta_2 = 0.999$) is used for training. $N$ is set to 15 in Fig. 1 (c). MDSF has two branches with filter kernel sizes of $3 \times 3$ and $5 \times 5$, respectively, and the number of groups is 8. We use PyTorch to implement our models on an NVIDIA Tesla V100 GPU.

## 4.2 MAIN RESULTS

**Single-image defocus deblurring results.** Tab. 1 shows comparisons of defocus deblurring methods on DPDD (Abuolaim & Brown, 2020). Our SFNet surpasses other state-of-the-art methods on most metrics. Particularly on the combined scene, SFNet obtains 0.25 dB PSNR improvement compared to the strong Transformer method Restormer (Zamir et al., 2022). In addition, our method provides a significant gain of 0.51 dB PSNR over the pre-trained network DRBNet (Ruan et al., 2022). The visual results in Fig. 2 show that our method recovers more details than other algorithms. Due to space limitations, experimental results of the dual-pixel setting are provided in Appendix E.

Figure 3: Image dehazing results on the SOTS-Indoor dataset (Li et al., 2018).

Table 2: Image dehazing comparisons on the synthetic dehazing datasets: SOTS-Outdoor and SOTS-Indoor (Li et al., 2018).

| Method | SOTS-Indoor PSNR↑ | SOTS-Indoor SSIM↑ | SOTS-Outdoor PSNR↑ | SOTS-Outdoor SSIM↑ |
|---|---|---|---|---|
| DehazeNet | 19.82 | 0.821 | 24.75 | 0.927 |
| AOD-Net | 20.51 | 0.816 | 24.14 | 0.920 |
| GridDehazeNet | 32.16 | 0.984 | 30.86 | 0.982 |
| MSBDN | 33.67 | 0.985 | 33.48 | 0.982 |
| FFA-Net | 36.39 | 0.989 | 33.57 | 0.984 |
| ACER-Net | 37.17 | 0.990 |  | - |
| DeHamer | 36.63 | 0.988 | 35.18 | 0.986 |
| MAXIM-2S | 38.11 | 0.991 | 34.19 | 0.985 |
| PMNet | 38.41 | 0.990 | 34.74 | 0.985 |
| DehazeFormer-L | 40.05 | **0.996** |  | - |
| **SFNet** | **41.24** | **0.996** | **40.05** | **0.996** |

Table 3: Image motion deblurring results on GoPro (Nah et al., 2017) and HIDE (Shen et al., 2020) datasets.

| Method | GoPro PSNR↑ | GoPro SSIM↑ | HIDE PSNR↑ | HIDE SSIM↑ |
|---|---|---|---|---|
| DeblurGAN-v2 | 29.55 | 0.934 | 26.61 | 0.875 |
| DBGAN | 31.10 | 0.942 | 28.94 | 0.915 |
| DMPHN | 31.20 | 0.940 | 29.09 | 0.924 |
| SPAIR | 32.06 | 0.953 | 30.29 | 0.931 |
| MIMO-UNet+ | 32.45 | 0.957 | 29.99 | 0.930 |
| IPT | 32.52 | - |  | - |
| MPRNet | 32.66 | 0.959 | 30.96 | 0.939 |
| HINet | 32.71 | 0.959 | 30.32 | 0.932 |
| Restormer | 32.92 | 0.961 | **31.22** | **0.942** |
| Stripformer | 33.08 | 0.962 | 31.03 | 0.940 |
| **SFNet** | **33.27** | **0.963** | 31.10 | 0.941 |

Table 4: Image dehazing results on Dense-Haze dataset (Ancuti et al., 2019).

| Method | PSNR↑ | SSIM↑ |
|---|---|---|
| DehazeNet | 9.48 | 0.438 |
| GridDehazeNet | 14.96 | 0.533 |
| FFA-Net | 12.22 | 0.444 |
| MSBDN | 15.13 | 0.555 |
| DeHamer | 16.62 | 0.560 |
| **SFNet** | **17.46** | **0.578** |

Table 5: Image motion deblurring results on RSBlur dataset (Rim et al., 2022).

| Method | PSNR↑ | SSIM↑ |
|---|---|---|
| SRN-DeblurNet | 32.53 | 0.840 |
| MIMO-UNet+ | 33.37 | 0.856 |
| MPRNet | 33.61 | 0.861 |
| Restormer | 33.69 | 0.863 |
| Uformer-B | 33.98 | 0.866 |
| **SFNet** | **34.35** | **0.872** |

Table 6: Image desnowing results on CSD (2000) dataset (Chen et al., 2021b).

| Method | PSNR↑ | SSIM↑ |
|---|---|---|
| DesnowNet | 20.13 | 0.81 |
| JSTASR | 27.96 | 0.88 |
| HDCW-Net | 29.06 | 0.91 |
| TransWeather | 31.76 | 0.93 |
| MSP-Former | 33.75 | 0.96 |
| **SFNet** | **38.41** | **0.99** |

**Image dehazing results.** We perform dehazing experiments on the synthetic benchmark RESIDE (Li et al., 2018) and real hazy dataset Dense-Haze (Ancuti et al., 2019). For RESIDE, we train our models for the indoor and outdoor scenarios separately, and then evaluate on the corresponding SOTS-Indoor and SOTS-Outdoor testsets. The quantitative results are shown in Tab. 2. Our method obtains the highest scores on all metrics. Particularly on the outdoor scene, our network generates a substantial gain of 4.87 dB PSNR over DeHamer (Guo et al., 2022). Compared to the recent works, DehazeFormer-L (Song et al., 2022) and PMNet (Ye et al., 2022), our method receives 0.19 dB and 2.83 dB higher PSNR on SOTS-Indoor testset, respectively. Additionally, we validate the performance of our approach on the real hazy dataset Dense-Haze (Ancuti et al., 2019). The results are shown in Tab. 4. As we can see, SFNet exhibits the superior ability of dealing with the real-world dehazing problem, receiving a gain of 0.84 dB over DeHamer (Guo et al., 2022). The dehazed results in Fig. 3 illustrate that SFNet is more effective in removing haze than other methods.

**Image motion deblurring results.** We evaluate our method on both the synthetic and real-world datasets. The numerical comparisons on the synthetic GoPro (Nah et al., 2017) and HIDE (Shen et al., 2020) datasets are reported in Tab. 3. On GoPro, SFNet shows 0.35 dB PSNR performance improvement over Restormer (Zamir et al., 2022) with ∼3× faster inference speed (Tab. 8). Moreover, our method receives a performance gain of 0.19 dB compared to the recent algorithm Stripformer (Tsai et al., 2022) with 34% fewer parameters and 26% fewer FLOPs. Notably, our SFNet

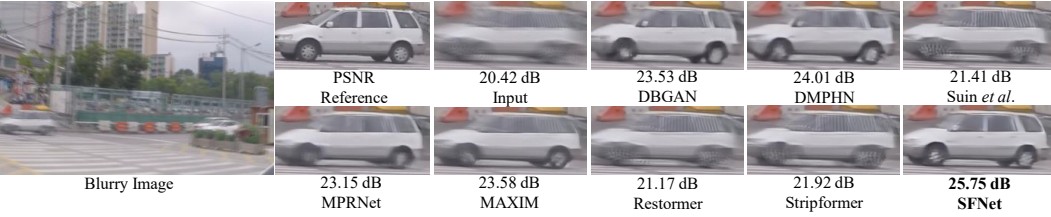

Figure 4: Image motion deblurring results on the GoPro dataset (Nah et al., 2017).

Table 7: Deraining comparisons with previous methods on five deraining datasets: Rain100H (Yang et al., 2017), Rain100L (Yang et al., 2017), Test100 (Zhang et al., 2019a), Test1200 (Zhang & Patel, 2018) and Test2800 (Fu et al., 2017).

| | Test100 | | Rain100H | | Rain100L | | Test2800 | | Test1200 | | Average | |
|---|---|---|---|---|---|---|---|---|---|---|---|---|
| Method | PSNR | SSIM | PSNR | SSIM | PSNR | SSIM | PSNR | SSIM | PSNR | SSIM | PSNR | SSIM |
| DerainNet | 22.77 | 0.810 | 14.92 | 0.592 | 27.03 | 0.884 | 24.31 | 0.861 | 23.38 | 0.835 | 22.48 | 0.796 |
| SEMI | 22.35 | 0.788 | 16.56 | 0.486 | 25.03 | 0.842 | 24.43 | 0.782 | 26.05 | 0.822 | 22.88 | 0.744 |
| DIDMDN | 22.56 | 0.818 | 17.35 | 0.524 | 25.23 | 0.741 | 28.13 | 0.867 | 29.65 | 0.901 | 24.58 | 0.770 |
| UMRL | 24.41 | 0.829 | 26.01 | 0.832 | 29.18 | 0.923 | 29.97 | 0.905 | 30.55 | 0.910 | 28.02 | 0.880 |
| RESCAN | 25.00 | 0.835 | 26.36 | 0.786 | 29.80 | 0.881 | 31.29 | 0.904 | 30.51 | 0.882 | 28.59 | 0.857 |
| PreNet | 24.81 | 0.851 | 26.77 | 0.858 | 32.44 | 0.950 | 31.75 | 0.916 | 31.36 | 0.911 | 29.42 | 0.897 |
| MSPFN | 27.50 | 0.876 | 28.66 | 0.860 | 32.40 | 0.933 | 32.82 | 0.930 | 32.39 | 0.916 | 30.75 | 0.903 |
| MAXIM-2S | 31.17 | **0.922** | 30.81 | 0.903 | 38.06 | **0.977** | 33.80 | **0.943** | 32.37 | **0.922** | 33.24 | **0.933** |
| **SFNet** | **31.47** | 0.919 | **31.90** | **0.908** | **38.21** | 0.974 | 33.69 | 0.937 | **32.55** | 0.911 | **33.56** | 0.929 |

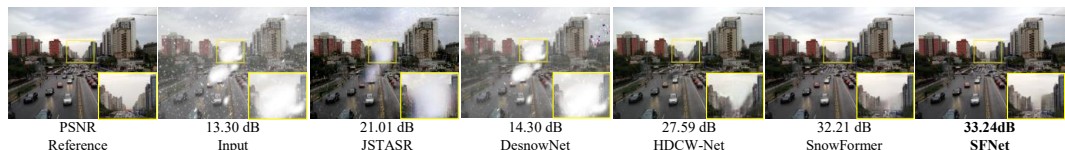

Figure 5: Image desnowing results on the CSD dataset (Chen et al., 2021c).

demonstrates stronger generalization capability to HIDE dataset than Stripformer on all metrics. In addition to the synthetic datasets, we further evaluate the effectiveness of our network on the real-world dataset. Tab. 5 shows quantitative comparisons on the newly proposed RSBlur (Rim et al., 2022) dataset. SFNet sets new state-of-the-art on this dataset, providing a substantial gain of 0.37 dB PSNR over the previous best method Uformer-B (Wang et al., 2022). Fig. 4 illustrates that SFNet produces more visually pleasant result than competing algorithms.

**Image desnowing results.** We compare our method on the CSD (Chen et al., 2021b) dataset with existing state-of-the-art methods (Chen et al., 2022a; Valanarasu et al., 2022; Chen et al., 2022c). As shown in Tab. 6, our framework yields a 4.66 dB PSNR improvement over the Transformer model MSP-Former (Chen et al., 2022c). The visual results in Fig. 5 show that our method is more effective in removing spatially varying snowflakes than competitors. More results on SRRS (Chen et al., 2020a) and Snow100K (Liu et al., 2018) are provided in Appendix E.

**Image deraining results.** Following recent works (Jiang et al., 2020; Purohit et al., 2021; Tu et al., 2022), we compare PSNR/SSIM scores on the Y channel in YCbCr color space. Tab. 7 shows that our method achieves the best performance on the average PSNR category compared to competing approaches. Moreover, on the Test100 dataset (Zhang et al., 2019a), the proposed SFNet obtains a performance boost of 0.30 dB PSNR over MLP model MAXIM-2S (Tu et al., 2022). Visual results shown in Fig. 6 illustrate that our model recovers more fine details without artifacts.

**Computational overhead comparisons.** In Tab. 8, we evaluate the computational costs of five motion deblurring methods on the GoPro testset (Nah et al., 2017). Evaluated on the full-resolution image, our method achieves fastest speed than other state-of-the-art algorithms while achieving comparable performance with fewer parameters.

Table 8: Overall comparisons between motion deblurring methods on the GoPro (Nah et al., 2017) test set.

| Method | NAFNet | MPRNet | DeepRFT+ | Restormer | SFNet |
|---|---|---|---|---|---|
| PSNR | **33.63** | 32.66 | 33.23 | 32.92 | 33.27 |
| Time/s | 0.833 | 1.148 | 0.806 | 1.218 | **0.408** |
| Params/M | 67.8 | 20.1 | 23.0 | 26.13 | **13.27** |
| FLOPs/B | **63.33** | 777.01 | 187.04 | 140.99 | 125.43 |

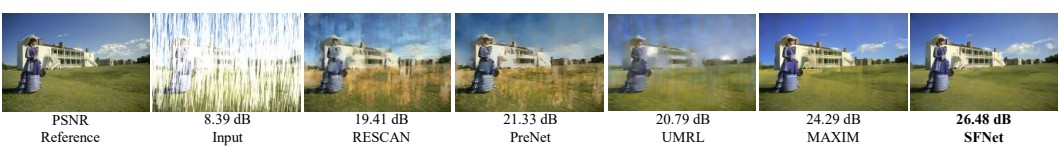

Figure 6: Image deraining results on the Rain100H dataset (Yang et al., 2017).

Table 9: Ablation studies for individual proposed modules.

| Method | PSNR | Params (M) | FLOPs (G) |
|---|---|---|---|
| Baseline | 31.20 | 6.90 | 66.32 |
| MDSF | 31.42 | 7.04 | 66.55 |
| MCSF | 31.45 | 6.92 | 66.38 |
| Full | 31.68 | 7.05 | 66.61 |

Table 10: Ablation study for the number of MCSF.

| Number | PSNR | Params | FLOPs |
|---|---|---|---|
| 2 | 31.22 | 6.91 | 66.34 |
| 4 | 31.33 | 6.91 | 66.35 |
| 6 | 31.42 | 6.92 | 66.37 |
| 8 | 31.45 | 6.92 | 66.38 |

Table 11: Ablation studies for MDSF.

| Group | PSNR |
|---|---|
| 2 | 31.40 |
| 4 | 31.40 |
| 8 | 31.42 |
| 16 | 31.30 |

## 4.3 ABLATION STUDIES

In this section, we first demonstrate the effectiveness of the proposed modules, and then investigate the effects of different designs for each module. Finally, we delve into the mechanism of MDCF to demonstrate its validity. Following the recent method (Tu et al., 2022), all models are trained on the GoPro (Nah et al., 2017) dataset for 1000 epochs, and $N$ is set to 7 in Fig. 1.

**Influence of each module.** Tab. 9 shows that MDSF and MCSF yield performance gains of 0.22 and 0.25 dB over the baseline model with few introduced computing burdens. Deployed only in a single position in each scale, MDSF produces the similar performance with MCSF, demonstrating the effectiveness of dynamic frequency selection mechanism. Furthermore, in Fig. 7, we plot the statistic differences between the ground truth and results of three methods on dehazing. With the frequency selection mechanism, the statistics of our results are closer to that of ground truth.

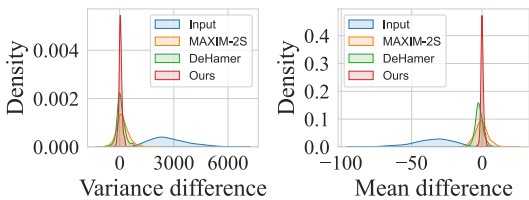

Figure 7: Variance (Left) and Mean (Right) difference between the ground truth and input/results of three methods on SOTS-Outdoor testset. Models are trained on OTS dataset (Li et al., 2018).

**Design choices for MCDF.** We study the influence of the number of MCDF in Tab. 10, where 2 MCSF means that we employ the proposed MCSF in last two residual blocks of each ResBlock. As can be seen, using more MCSF leads to the consistently increasing performance from 31.22 to 31.45 dB PSNR while only introducing 0.01 M parameters and 0.04 G FLOPs. Due to its few introduced parameters and low complexity, we insert MCSF in each residual block for frequency learning.

**Design choices for MDSF.** To understand the impact of the number of groups in MDSF, we test various configurations in Tab. 11. Generally, the increasing number of groups leads to higher PSNR, demonstrating the effectiveness of the filter diversity. However, the accuracy saturates at group 16, which is probably caused by overfitting. We finally pick 8 groups for better performance.

**Alternatives for MDSF.** To examine the advantage of our design, we compare our decoupler with several alternatives in Tab. 12. We first substitute the learning-based and fixed frequency separation methods for our decoupler. We form *Conv* method (Tab. 12a) by using strided convolution to generate different frequency parts with reduced resolution (Pang et al., 2020). The *Oct-conv* (Tab. 12b) version (Chen et al., 2020b) shares the similar idea with *Conv*, which utilizes down-sampling to reduce network redundancy.

Table 12: Alternatives for MDSF.

| Method | PSNR | Params (M) | FLOPs (B) |
|---|---|---|---|
| (a) Conv | 31.11 | 8.65 | 79.35 |
| (b) OctConv | 31.07 | 6.99 | 66.86 |
| (c) Gaussian | 30.98 | 7.02 | 66.81 |
| (d) Wavelet | 30.97 | 7.06 | 66.33 |
| (e) Window | 31.22 | 6.98 | 66.73 |
| (f) Local | 31.23 | 7.00 | 66.34 |
| (g) MDSF | **31.42** | 7.04 | 66.55 |

These variants only introduce extra low-frequency signal to the network. We further utilize fixed separation methods to replace the proposed decoupler. *Gaussian* (Tab. 12c) and *Wavelet* (Tab. 12d) produce the similar results, much lower than our MDSF. Additionally, *Wavelet* needs more parameters to deal with its multiple branches.

Since our filter kernel is generated by learning, we further compare our MDSF with two attention approaches to verify the validity of the proposed selection mechanism. Specifically, we utilize the widely used window-based self-attention (Wang et al., 2022) (Tab. 12e) and dynamic convolution (Han et al., 2021; Wu et al., 2019) (Tab. 12f) to conduct comparisons. As we can see from the table, our method has huge advantages over these methods, demonstrating the effectiveness of MDSF.

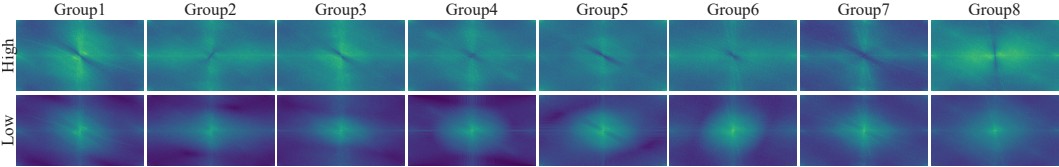

Figure 8: The variance and discrete Fourier transform of the resulting images as we iteratively impose the produced filters on the image. **Left:** The low-pass filter. **Right:** The high-pass filter.

Figure 9: Discrete Fourier transform results of group-wise features generated by our MDSF decoupler. The results are sampled from the last decoder. **Top:** High-frequency. **Bottom:** Low-frequency.

## 4.4 QUALITATIVE ANALYSES OF MDSF

We provide qualitative analyses of MDSF based on discrete Fourier transform. Results are obtained from the branch of $3 \times 3$ filter. The input image is sampled from GoPro (Nah et al., 2017), shown in Fig. 10. The features are obtained from the last residual block in the last ResBlock of decoder.

We first verify the properties of the alleged low-/high-pass filters in MDSF. To this end, we iteratively apply the produced filters to the image. The variance and corresponding spectral features of intermediate images are provided in Fig. 8. Taking the low-pass filter as an example, with the increasing of iteration times, the variance of the image decreases constantly, and the high-frequency signals in spectral features are reduced drastically. The high-pass filter exhibits the opposite properties. These results demonstrate the effectiveness of our filters. It is remarkable that the high-pass filter produces large variance with fewer iterations, hence it is more effective than the low-pass filter. As a result, it is easy for MDSF to introduce more high-frequency signals into the network for reconstruction.

In MDSF, we generate different filters for each group to enhance the diversity of frequency features. To delve into this mechanism, we visualize the group-wise spectral features in Fig. 9. As expected, different groups focus on the learning of disparate low-/high-frequency signals, enriching the diversity of frequency representations for selection. We further compare the feature maps before and after our MDSF in Fig. 10. Using the attained filters, the decoupler of MDSF produces different frequency components. The high-frequency feature contains much edge information. The resulting feature after modulator recovers more details of the number plate that is blurry in the initial feature.

## 5 CONCLUSION

We present an image restoration framework, SFNet, which is built on frequency selection mechanism. We develop two key modules, MDSF and MCSF, to conduct frequency decomposition and recalibration with different receptive fields. Specifically, our multi-branch dynamic selective frequency module (MDSF) builds a dynamic filter to decompose feature maps into various frequency parts and utilizes channel attention to perform accentuation, thus effectively selecting the most informative frequency to recover. Furthermore, the proposed multi-branch compact selective frequency module (MCSF) introduces a simple yet effective manner to enlarge the receptive field and conduct frequency selection. With both designed modules, SFNet achieves state-of-the-art results on five image restoration tasks, demonstrating the validity of our frequency selection mechanism.

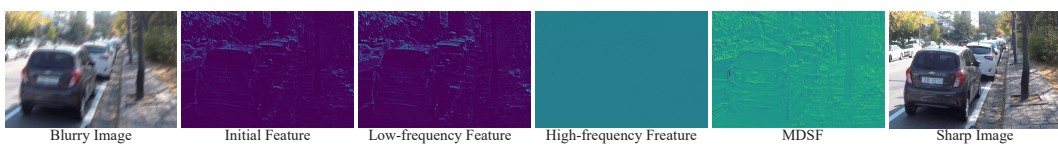

Figure 10: The internal features of MDSF. With our frequency selection mechanism, MDSF produces more fine details than the initial feature, *e.g.,* the number plate. Zoom in for the best view.

ACKNOWLEDGEMENT

This work was supported by the National Natural Science Foundation of China under Grant (62172409), Shenzhen Science and Technology Program (JCYJ20220818102012025, JCYJ20220530145209022), and Beijing Nova Program (Z201100006820074).

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
