# OpenReview forum: "Selective Frequency Network for Image Restoration"
_ICLR.cc/2023/Conference — ICLR 2023 poster_

### Official Review · Reviewer_Qxbc · 2022-10-24

**Confidence:** 5
**Correctness:** 4
**Technical Novelty And Significance:** 3
**Empirical Novelty And Significance:** 3
**Recommendation:** 8

**Clarity, Quality, Novelty And Reproducibility:**

The paper is novel enough for its proposed modules MDSF and MCSF as well as SFN established with them. The content is presented clearly and organized reasonably, and the technique described could be reproduced easily. Lastly, the writing is good and fluent.

**Strength And Weaknesses:**

The paper pays attention to the large discrepancy of sharp and degraded image pairs in the frequency domain, and proposes the modules MDSF and MCSF which are used to build SFN net. It is novel in terms of method. The structure of the proposed SFN is clearly presented in the text, and more details are further discussed in Appendex A-H. The content of the paper is very sufficient. The proposed method is validated by extensive experiments to be effective on many image restoration tasks, and perform favorably compared with lots of SOTS methods on various datasets. In addition, ablation studies are made to explain the design of the modules MDSF and MCSF.

1.What are the advantages of the “Shallow Layer and output layer” over existing multi-scale methods?
2.Why MDSF and MCSF are concatenated in Figure 1? These two modules are used independently, right?
3.How to control the kernel size of different decouplers in MDSF? From figure 2(b), it’s seems that all the kernel size are k^2?
4. There are still some typos in the paper:
(1) In the last paragraph of Section 2, both “Xu et al.” and “Mao et al.” lack reference numbers.
(2) In Figure 2(b), the output of MDSF is the concatenation of the low-frequency feature and the high-frequency feature, while in Figure 2(c), MCSF receives two inputs, i.e., the low-frequency feature and the high-frequency feature from MDSF. I wonder if the concatenation operation in Figure 2(c) is necessary.
(3) The parameter c in equations (2) and (4) takes a real value as in c \in R^{C_i} and c\in R^{2C}. However, c is the index of a channel according to the context of the paper, and could take an integer value.
(4) Section 3.2.1 could be modified to Section 3.3.

**Summary Of The Paper:**

The paper proposes a new deep model framework named Selective Frequency Network (SFN) for multiple image restoration tasks, including single image defocus deblurring, image dehazing, image motion deblurring, image desnowing and image deraining. SFN builds on two types of basic blocks, i.e., Multi-branch Dynamic Selective Frequency module (MDSF) and Multi-branch Compact Selective Frequency module (MCSF). MDSF is proposed to decompose the feature into frequency subbands dynamically and locally for accentuating the effective ones further, while MCSF is designed by using global and window-based average pooling to deal with the large-scale degradation kernel. The proposed SFN exploits a U-type structure to assemble the modules of MDSF and MCSF. Extensive experiments are carried out to validate the superiority of SFN and the effectiveness of block components MDSF and MCSF.

**Summary Of The Review:**

The paper is novel enough for its proposed modules MDSF and MCSF as well as SFN established with them. The content is presented clearly and organized reasonably, and the technique described could be reproduced easily. Lastly, the writing is good and fluent.

---

> ### Author Response · Authors · 2022-11-18
> **Response to Reviewer Qxbc**
>
> Dear Reviewer Qxbc,
>
> We would like to sincerely thank you for your detailed review and positive feedback.  We will elaborately address your concerns and questions as follow.
>
> >Q 1: What are the advantages of the “Shallow Layer and output layer” over existing multi-scale methods?
>
> A 1:  The shallow layer and output layer incorporate multi-scale input and output to ease the training difficulty. They share the same idea with the multi-resolution input/output operation in the multi-scale networks, but implement in an efficient manner without using multiple stacked U-type networks. These layers are suitable for the framework with a single U-Net backbone.
>
> >Q 2: Why MDSF and MCSF are concatenated in Figure 1? These two modules are used independently, right?
>
> A 2: Yes, two modules can be used independently. Inspired from the multi-patch method DMPHN [1]  that learns fine-to-coarse hierarchical representations, we simply integrate two modules in the order of MDSF and MCSF, where MDSF focuses on local information and MCSF enlarges the receptive field. Recently, we have tried more integration methods as shown in the table.
> | Method | PSNR |
> |-------|---------|
> |MDSF-MCSF|31.68|
> |MCSF-MDSF|31.68|
> |Parallel|31.73|
>
> As we can see, the coarse-to-fine variant (MCSF-MDSF) achieves the same performance with our version. Interestingly, the parallel scheme outperforms two cascaded variants.  These experiments demonstrate the effectiveness of our modules. We will investigate the better combination method to excavate the potential of our designs in the future.
>
>
> >Q 3: How to control the kernel size of different decouplers in MDSF? From figure 2(b), it’s seems that all the kernel size are k^2?
>
> A 3: We assign different k for each decoupler. Despite the enlarged receptive field, large kernel size will introduce more complexities. To strike a balance between accuracy and computation overhead, we simply choose k=3 and k=5 for the two decouplers in the final network.
>
> >Q 4: There are still some typos in the paper: (1) In the last paragraph of Section 2, both “Xu et al.” and “Mao et al.” lack reference numbers. (2) In Figure 2(b), the output of MDSF is the concatenation of the low-frequency feature and the high-frequency feature, while in Figure 2(c), MCSF receives two inputs, i.e., the low-frequency feature and the high-frequency feature from MDSF. I wonder if the concatenation operation in Figure 2(c) is necessary. (3) The parameter c in equations (2) and (4) takes a real value as in c \in R^{C_i}   and c\in R^{2C}. However, c is the index of a channel according to the context of the paper, and could take an integer value. (4) Section 3.2.1 could be modified to Section 3.3.
>
> A 4:  Thanks for pointing out our minor typos. We have carefully polished our draft.
>
> (1) We have added the corresponding citations in Sec. 2.
>
> (2) We have refined our figure for clarity.  In the initial version, the concatenation in decoupler is used for the group-wise feature maps within a multi-channel frequency feature while the concatenation in modulator is to concatenate different frequency features.
>
> (3) Yes, $c$ is the index of a channel. We have revised it.
>
> (4) We have corrected that in the revision.
>
>
> Reference
>
> [1] Hongguang Zhang, Yuchao Dai, Hongdong Li, and Piotr Koniusz. Deep Stacked Hierarchical Multi-Patch Network for Image Deblurring. In CVPR 2019.

---

### Official Review · Reviewer_2iAU · 2022-10-24

**Confidence:** 4
**Correctness:** 3
**Technical Novelty And Significance:** 4
**Empirical Novelty And Significance:** 3
**Recommendation:** 8

**Clarity, Quality, Novelty And Reproducibility:**

The manuscript is basically clear. The technical content of the paper is correct. The present paper is reproducible. Incorporating the proposed MDSF and MCSF into the selective frequency network is able to address most image restoration tasks.

**Strength And Weaknesses:**

Strengths:
1. The authors design a useful module to decouple the feature map into different parts and select the most important ones for restoration.
2. The network architecture is well-introduced and it is easy to reproduce the main results.
3. The experiments show that the proposed network achieves state-of-the-art results on five image restoration tasks. The supplementary material shows more results to demonstrate the effectiveness of the proposed method.

Weaknesses:
1. It will be better if the authors can integrate and simplify Figures 1-3 properly.
2. How to switch the high-pass and low-pass filters in the Filter Switcher automatically?
3. More visual results or the corresponding PSNRs can be added in Figure 5 to show the superior of the proposed network. Besides, the typesetting of Figure 5 should be modified. If the space is limited, try to delete some compared results in Table1-7.
4. Are the ablation studies all about the deblurring task? If so, add this explanation.
5. The FLOPs should be added in Table 10 to support that more MCDF modules bring few parameters and complexities.


**Summary Of The Paper:**

This paper focuses on image restoration tasks in the frequency domain. Specially, a multi-branch dynamic selective frequency module is employed to select the most informative frequency to recover. Besides, the authors propose a multi-branch compact selective frequency module to enlarge the receptive field. Experimental results demonstrate that the network utilizing these two modules performs better than state-of-the-art methods on different image restoration tasks.

**Summary Of The Review:**

I tend to accept this paper based on its solid contributions and novelty.

---

> ### Author Response · Authors · 2022-11-18
> **Response to Reviewer 2iAU**
>
> Dear Reviewer 2iAU,
>
> Thanks for your valuable comments and recognition of our contribution.  We highly appreciate your suggestions, and believe they helped improve the quality of our paper. Please find our detailed response below.
>
> >Q 1: It will be better if the authors can integrate and simplify Figures 1-3 properly.
>
> A 1:  We thank the reviewer for improving our paper. We have redrawn the figures in the revision.
> >Q 2:  How to switch the high-pass and low-pass filters in the Filter Switcher automatically?
>
> A 2: The Filter Switcher is used for different filters. We clarify that our module generates both high-/low-frequency signals simultaneously, and we removed that Switcher in the revision to avoid misunderstanding.
> >Q 3:  More visual results or the corresponding PSNRs can be added in Figure 5 to show the superior of the proposed network. Besides, the typesetting of Figure 5 should be modified. If the space is limited, try to delete some compared results in Table1-7.
>
> A 3:  Thanks for the valuable comments. We have updated the visual results in Fig 2-6, added corresponding PSNRs for each image, and adjusted the typesetting of the figure.
>
> >Q 4:  Are the ablation studies all about the deblurring task? If so, add this explanation.
>
> A 4:  Yes, all about the deblurring task following the recent algorithm MAXIM [1]. We have added the explanation in Sec. 4.3.
> >Q 5:  The FLOPs should be added in Table 10 to support that more MCDF modules bring few parameters and complexities.
> >
> A 5:  Thanks for your suggestion. We have provided the detailed parameters and FLOPs for different numbers of MCDF in Tab. 10.  With six more MCDF modules, the performance is improved from 31.22 to 31.45 dB PSNR by introducing only 0.01 M parameters and 0.04 G FLOPs.
>
> Reference
>
> [1] Zhengzhong Tu, Hossein Talebi, Han Zhang, Feng Yang, Peyman Milanfar, Alan Bovik, and Yinxiao Li. MAXIM: Multi-Axis MLP for Image Processing. In CVPR 2022.

---

> > ### Comment · Reviewer_2iAU · 2022-12-08
> > **Post-rebuttal**
> >
> > Thanks for your response. My concerns are well addressed in the response. I prefer to accept this paper.

---

> > > ### Author Response · Authors · 2022-12-08
> > > **Thank you for the response.**
> > >
> > > Dear Reviewer 2iAU,
> > >
> > > Thank you for the response. We are glad our responses were helpful in addressing your concerns. We really appreciate your time and insightful comments that led to significant improvement of our manuscript.
> > >
> > > Thank you again.
> > >
> > > Best regards,
> > >
> > > Authors of Paper6167

---

### Official Review · Reviewer_4vXg · 2022-10-24

**Confidence:** 4
**Correctness:** 2
**Technical Novelty And Significance:** 2
**Empirical Novelty And Significance:** 3
**Recommendation:** 6

**Clarity, Quality, Novelty And Reproducibility:**

-The paper quality needs to be improved.
-The authors should further refine the manuscript organization and writing, e.g., Section 3.2.1.
-The authors can refine and reorganize the figures, such as Fig. 2 and Fig.3, to make them clearer and more good-looking. Some other analysis figures are also blurry.
-Some citations are missing in the manuscript, e.g., Mao et al. in Frequency based image restoration, Section 2.

**Strength And Weaknesses:**

Strength
-The paper works on frequency decomposition and manipulation design, which is important for image restoration.
-The proposed method outperforms many state-of-the-art methods on different image restoration tasks.

Weakness
-Comparison with other attention mechanisms. The proposed MDSF and MCSF are similar to the multi-brach pathways with attention mechanisms. Since the kernel weights are learned, it is hard to tell whether the learned dynamic frequency decomposition benefits more than attention. The authors can replace the modules in Fig 2. (b) with other attention forms to verify the argument.
-Analysis of the high- and low-frequency features. The authors should provide more analysis of the obtained features of MSDF to verify the high- and low-frequency properties and frequency diversity.
-Ablation studies. The authors conduct ablation studies with different settings. It is hard to compare across different components. The authors should give a more unified ablation.
-Comparison to state-of-the-art. The authors should include more recent works, such as NAFNet for image debarring, in the comparison. It is also not obvious whether the proposed modules or training strategies contribute more to the final results.  Liangyu Chen, et al. "Simple baselines for image restoration." arXiv preprint arXiv:2204.04676 (2022).
-The authors can include other related works, e.g., Jaeyoung Yoo, Sang-ho Lee, and Nojun Kwak. "Image restoration by estimating frequency distribution of local patches." CVPR 2018
-The paper quality needs to be improved.

**Summary Of The Paper:**

This paper works on image restoration and introduces a multi-branch dynamic selective frequency (MDSF) module with learned kernels to generate high and low-frequency features and a multi-branch compact selective frequency (MCSF) module to enhance the features with an attention mechanism. The proposed method achieves several state-of-the-art performances for image restoration tasks.

**Summary Of The Review:**

The paper achieves good results for different image restoration tasks. However, the paper lacks some validation of the proposed method and comparison to more recent works. Also, the paper quality needs to be improved.

---

> ### Author Response · Authors · 2022-11-18
> **Response to Reviewer 4vXg**
>
> Dear Reviewer 4vXg,
>
> Thanks for your time and valuable feedback. We will elaborately address your concerns and questions as follows.
>
> >Q 1: Comparison with other attention mechanisms. The proposed MDSF and MCSF are similar to the multi-branch pathways with attention mechanisms. Since the kernel weights are learned, it is hard to tell whether the learned dynamic frequency decomposition benefits more than attention. The authors can replace the modules in Fig 2. (b) with other attention forms to verify the argument.
>
> A 1:  We utilize the learned and theoretically proved filter to decouple the feature map. Here we provide comparisons between our MDSF and two alternatives. The results are shown in the table.
>
> | Method | PSNR | FLOPs/G | Params/M|
> |-------|---------|---------|----------|
> |Window-based self-attention [1]|31.22|66.73|6.98|
> | Dynamic convolution [2]|31.23|66.34|7.00|
> |MDSF|31.42|66.55|7.04|
>
> As we can see, our MDSF performs better than the window-based attention with fewer FLOPs. The dynamic convolution whose kernel is also learned is inferior to our module.  We argue that our design focuses on the frequency selection mechanism that is orthogonal to the attention module design.
>
> Moreover, we can design an experiment by simply removing the decoupling mechanism in MCSF to verify our argument. The comparisons are shown in the table.
> | Method | PSNR |
> |-------|---------|
> |MCSF w/o decoupling|31.27|
> |MCSF|31.45|
>
> The provided experiments further demonstrate the effectiveness of our mechanism.
>
>
> >Q 2: Analysis of the high- and low-frequency features. The authors should provide more analysis of the obtained features of MSDF to verify the high- and low-frequency properties and frequency diversity.
>
> A 2:  Thanks for your suggestion. To better understand the property of our filters in MDSF, in Fig. 8 we provide the variation trend of variance and corresponding spectral features as we iteratively apply the generated filters to an image. We also provide the group-wise spectral features in Fig. 9 to verity our frequency diversity. Different groups focus on the learning of disparate low-/high-frequency signals. Furthermore, we exhibit the features before and after deploying our MDSF in Fig. 10. MDSF helps recover more details. We added the corresponding analyses in Section 4.4.
>
> >Q 3: Ablation studies. The authors conduct ablation studies with different settings. It is hard to compare across different components. The authors should give a more unified ablation.
>
> A 3:  Thanks for your helpful feedback. We have unified the training epochs, and detailed experimental settings in Sec. 4.3.
>
> >Q 4: Comparison to state-of-the-art. The authors should include more recent works, such as NAFNet for image debarring, in the comparison. It is also not obvious whether the proposed modules or training strategies contribute more to the final results.
>
> A 4:  Thank for you suggestion. We have included NAFNet [3] for the deblurring task in Tab. 8. Our model achieves the comparable performance with 80% fewer parameters and runs 2x faster than NAFNet.
> We introduce our training strategies for each task in Appendix A of the supplementary material and Sec 4.1 of the main text. We do not utilize any customized training strategy for our model, and just adopt the commonly used configurations of previous restoration approaches, such as training epochs and patch size.
> Furthermore, to verify the effectiveness of our method, we apply our modules to MIMO-UNet [4]. The results on image deblurring are shown in the table.
> |     | Metrics | GoPro |HIDE|
> |--------|-----|------|--------|
> |MIMO-UNet|PSNR (dB)/SSIM|31.73/0.951|29.28/0.921|
> |MIMI-UNet w/ our method|PSNR (dB)/SSIM|32.20/0.955|30.14/0.931|
>
> We can see that our frequency selection mechanism obtains a remarkable improvement of 0.47 dB on the GoPro dataset, demonstrating the effectiveness of our modules.
>
> >Q 5: The authors can include other related works, e.g., Jaeyoung Yoo, Sang-ho Lee, and Nojun Kwak. "Image restoration by estimating frequency distribution of local patches." CVPR 2018
> >
> A 5:  Thanks for your suggestion. We have added more related works in Sec 2.
> >Q 6: The paper quality needs to be improved. The authors should further refine the manuscript organization and writing, e.g., Section 3.2.1. The authors can refine and reorganize the figures, such as Fig. 2 and Fig.3, to make them clearer and more good-looking. Some other analysis figures are also blurry.Some citations are missing in the manuscript, e.g., Mao et al. in Frequency based image restoration, Section 2.
>
> A 6:  Thanks for your valuable advice. We have carefully polished the draft by refining the organization and writing, redrawing the figures, updating visual comparisons, and adding the omitted citations.

---

> > ### Author Response · Authors · 2022-11-18
> > **Reference**
> >
> > Reference
> >
> > [1] Zhendong Wang, et al. Uformer: A general u-shaped transformer for image restoration. In CVPR, 2022.
> >
> > [2] Qi Han, et al. On the Connection between Local Attention and Dynamic Depth-wise Convolution. In ICLR 2021.
> >
> > [3] Liangyu Chen, et al. NAFNet: Nonlinear Activation Free Network for Image Restoration. In ECCV, 2022.
> >
> > [4] Sung-Jin Cho, et al. Rethinking Coarse-To-Fine Approach in Single Image Deblurring. In ICCV, 2021.

---

> ### Comment · Reviewer_4vXg · 2022-12-08
> **Final Rating**
>
> The authors solve most of my concerns. And I perfer to accept this paper.

---

> > ### Author Response · Authors · 2022-12-08
> > **Thanks for the response.**
> >
> > Dear Reviewer 4vXg,
> >
> > Thank you for the response and your positive feedback on our paper. We really appreciate your time engaged in the review and rebuttal phase, and our work has been greatly improved through your constructive comments.
> >
> > Thanks!
> >
> > Best regards,
> > Authors of Paper6167

---

### Official Review · Reviewer_BBSS · 2022-10-24

**Confidence:** 4
**Clarity, Quality, Novelty And Reproducibility:** 1) There are some problems in the pro…
**Correctness:** 3
**Technical Novelty And Significance:** 3
**Empirical Novelty And Significance:** 3
**Recommendation:** 8

**Strength And Weaknesses:**

The paper is well presented and claims better performance on different tasks, while there are some problems in theoretical proof and experimental part.

**Summary Of The Paper:**

In this paper, the author proposes a Selective Frequency Network (SFNet) for image restoration. In SFNet, the author designs a multi-branch and content-aware module to decompose the feature into separate frequency sub-bands, and then uses the channel-wise attention mechanism to emphasize the useful information. In addition, to cope with the large-scale degradation kernel, the author proposes an decoupling and modulation module to enlarge the receptive field based on global and window-based average pooling.

**Summary Of The Review:**

The paper is well presented and claims better performance on different tasks, while there are some problems in theoretical proof and experimental part.

---

> ### Author Response · Authors · 2022-11-18
> **Response to Reviewer BBSS**
>
> Dear Reviewer BBSS,
>
> Many thanks for your valuable time and comments. We carefully address your concerns in detail.
> >Q 1.1: In Eq. 8 and Eq. 9, the author describes that the first row(diag(1,0,0,…,0)) in the Fourier domain is low frequency and the others are high frequency. According to this description, for a feature of size CxHxW, low frequency is the mean value in the spatial dimension and the size of low frequency is Cx1x1, which can be seen in Section 3.2 and Appendix C. This description is contrary to the commonly used low frequency and high frequency. In practice, the size of low frequency and high frequency are CxHxW and low frequency occupies a large part of information and high frequency is only in the edge area. In the two-dimensional Fourier domain (the data has been processed by fftshift), we generally believe that the low frequency is the region within the radius R, and other regions are the high frequency [1-3]. Therefore, the Eq. 8 and Eq. 9 may not be consistent with the actual situation.
>
> A 1.1: Yes, we consider the mean filter as the low frequency operator in our proof. We define the low-/high-frequency operators to provide conveniences for our subsequent proof, instead of meaning that our MDSF is a low-pass filter only if it is a mean filter. In other words, the mean value is a goal for the low-pass filter after using many times. But this does not mean that after using the low-pass filter once, we can get the mean value. Based on the operator, the size of low frequency is Cx1x1 in the frequency domain, but it is applicable to all spatial pixels in the spatial domain.  The size gap is bridged by the broadcast mechanism of programming language.
>
> We added the features produced by MDSF in Fig 10, and the resulting low-/high-frequency features are in accord with the properties of low/high frequency as you mentioned.
>
> The low/high frequency is a relative concept. Our starting point is that if we iteratively apply a low-pass filter to an image, the image will become very blurry and only very low frequency signals are remained. Hence, in our proof we set a specific bound and consider the lowest frequency (R=1) as the result of using low-pass filter for many times. We iteratively using MDSF for many times, and FFT results in Fig. 8 are consistent with our hypothesis.
>
>
> >Q 1.2: In other words, the author only provided theoretical proof of MDSF for extreme low frequency but did not provide theoretical proof of MDSF for other low frequency.
>
> A 1.2: Imposing a low-pass filter on an image for unlimited times, only very low frequency can be remained. As the number of iterations increase, other relatively low frequency signals will also be filtered.
>
> >Q 1.3: More importantly, the author did not demonstrate that the low frequency in the MDSF meets Eq. 8. The low frequency in the MDSF is obtained through learning, not the mean value of the feature.
>
> A 1.3: Our proof is established on the knowledge that if we iteratively apply a low-pass filter to an image infinitely, only very low frequency can be remained. Here, we conclude our proof briefly as follows:
> 1) We first define our low-/high-frequency operators where low-frequency operator is mean filter.
> 2) In Theorem 1 ,  the low-pass filter is defined as:
> the ratio between high-frequency and omni-frequency signals will approach 0, as we increase the iterations.
> 3) The result of our MDSF (applied for infinite times) meets the definition in Theorem 1. Hence, our MDSF is a low-pass filter.
>
> Regarding your concern, our MDSF is a low-pass filter if it meets the requirement in Theorem 1 instead of Eq. 8. The property in Eq.8 is the ultimate goal of our MDSF after being using many times.
>
> >Q 1.4: In addition, the size of low frequency is CxHxW, which is completely inconsistent with the low frequency scale defined by the author. Therefore, this proof is problematic and the author confused the definition of low frequency.
>
> A 1.4: As we mentioned in <font color="#006666">A 1.1</font>, our low frequency scale is Cx1x1 in the frequency domain, and it is applicable to all pixels of size CxHxW in the spatial domain. The size is automatically unified by the broadcast mechanism of programming language.
>
> >Q 2: The author did not write clearly the experimental settings, such as the number of branches in MDSF, the number of groups in MDSF.
>
> A 2: We give detailed experimental settings here, and we have added them in the updated version (Sec. 4.1). MDSF has two branches and the number of groups in MDSF is experimentally set as 8.
> >Q 3: The experiment is not rigorous. In the ablation experiment, the number of epochs is inconsistent in the ablation experiments of different components. Authors should unify the number of epochs.
>
> A 3:  Thanks for your suggestion. We have unified the number of epochs for all ablation experiments.
> >Q 4: There are some typos in Section 4.8 and Eq. 15.
>
> A 4:  Thanks for pointing out minor typos. We have addressed them carefully in the revision.

---

> > ### Comment · Reviewer_BBSS · 2022-12-09
> > **The author address most of my concerns.**
> >
> > The author address most of my concerns. I perfer to accept this paper.

---

> > > ### Author Response · Authors · 2022-12-09
> > > **Thanks for the response.**
> > >
> > > Dear Reviewer BBSS,
> > >
> > > Thanks for your positive feedback, and we are highly encouraged to hear that our responses helped address your concerns. We would appreciate it if you could kindly adjust the score correspondingly. Thanks very much for your valuable time.
> > >
> > > Best regards,
> > >
> > > Authors of Paper6167

---

> ### Author Response · Authors · 2022-12-09
> **Gentle Reminder**
>
> Dear Reviewer BBSS,
>
> We greatly appreciate your time and constructive comments, and we have made every effort to faithfully answer all your comments in the response and manuscript. As we are approaching the end of the discussion period, we would appreciate it if you help us review our response and kindly let us know whether our explanations address your concerns. Thanks for your valuable time.
>
>
> Best regards,
>
> Authors of Paper6167

---

### Author Response · Authors · 2022-11-18
**General Response**

We thank all reviewers for their time and valuable feedback.  We responded to the individual comments of each reviewer. Additionally, we revised our manuscript with the key modifications marked in blue. The updates are listed as follows:

- We have refined the framework figure (Fig. 1) and adjusted the organization by adding content about the overall pipeline (Sec. 3.1).
- We have discussed more related works (Sec. 2).
- We have updated the figures of visual results for five tasks with corresponding PSNR (Fig. 2-6).
- We have provided the comparisons between our method and several attention mechanisms (Tab. 12).
- We have unified and specified the experimental settings for ablation studies (Sec. 4.3, Tab. 9-12).
- We have provided qualitative analyses for our MDSF (Fig. 8-10, Sec. 4.4).
- We have corrected the symbols and typos, and carefully polished our paper.

Finally, we thank all reviewers again for their thoughtful reviews.

Best regards, Authors of Paper6167

---

### Decision · Program_Chairs · 2023-01-20

**Decision:**

Accept: poster

**Justification For Why Not Higher Score:**

This paper is a novel and solid application work for image restoration tasks. Considering that it doesn't provide a fundamental contribution or make a breakthrough in the field, the AC recommends "accept with poster".

**Justification For Why Not Lower Score:**

The proposed method is novel and obtains state-of-the-art results on different image restoration tasks. The AC recommends accepting it.

**Metareview: Summary, Strengths And Weaknesses:**

The paper proposes two new modules, i.e., the multi-branch dynamic selective frequency module (MDSF) and the multi-branch compact selective frequency module (MCSF), and incorporates them into a U-shaped backbone for image restoration. Experiments are conducted to validate the effectiveness of the framework. The proposed method is novel and obtains state-of-the-art results on different image restoration tasks. After the first round of review, Reviewer BBSS pointed out that there were some issues in the proof. Reviewer 4vXg mentioned that the paper lacked some validation of the proposed method and comparison with more recent works.  Reviewer 2iAU provided some suggestions about how to improve the current presentation and experiments. Reviewer Qxbc asked some technical questions about the proposed method. During the rebuttal, the authors provide detailed replies and extra experiment results to address the above issues. The revision has been made in the paper by taking into account all valuable suggestions provided by the reviewers. The paper eventually receives 2 accepts, 1 marginal accept, and 1 marginal reject.  After reading the rebuttal and having an internal discussion, the AC thinks that all major concerns have been addressed by in the rebuttal and recommends accepting the paper due to its novelty and solid experiments.


**Note From Pc:**

if the above contains the word "oral" or "spotlight" please see: "oral" presentation means -> notable-top-5% and "spotlight" means -> notable-top-25%. As stated in our emails, we are disassociating presentation type from AC recommendations